# A bacterial type III effector hijacks plant ubiquitin proteases to evade degradation

Wenjia Yu[1,2☯], Meng Li[1,2☯], Wenjun Wang[1,2], Haiyan Zhuang[1], Jiamin Luo[1,2], Yuying Sang[1], Cecile Segonzac[3], Alberto P. Macho[1]*

1 Shanghai Center for Plant Stress Biology, CAS Center for Excellence in Molecular Plant Sciences, Chinese Academy of Sciences, Shanghai, China, 2 University of the Chinese Academy of Sciences, Beijing, China, 3 Department of Agriculture, Forestry and Bioresources, Seoul National University, Seoul, Republic of Korea

☯ These authors contributed equally to this work.
* alberto.macho@cemps.ac.cn, alberto.macho@icloud.com

## Abstract

Gram-negative bacterial pathogens inject effector proteins inside plant cells using a type III secretion system. These effectors manipulate plant cellular functions and suppress the plant immune system in order to promote bacterial proliferation. Despite the fact that bacterial effectors are exogenous threatening proteins potentially exposed to the protein degradation systems inside plant cells, effectors are relative stable and able to perform their virulence functions. In this work, we found that RipE1, an effector protein secreted by the bacterial wilt pathogen, *Ralstonia solanacearum*, undergoes phosphorylation of specific residues inside plant cells, and this promotes its stability. Moreover, RipE1 associates with plant ubiquitin proteases, which contribute to RipE1 deubiquitination and stabilization. The absence of those specific phosphorylation sites or specific host ubiquitin proteases leads to a substantial decrease in RipE1 protein accumulation, indicating that RipE1 hijacks plant post-translational modification regulators in order to promote its own stability. These results suggest that effector stability or degradation in plant cells constitute another molecular event subject to co-evolution between plants and pathogens.

## Author summary

Most bacterial plant pathogens inject effector proteins inside host cells to suppress immune responses and manipulate other plant functions in order to cause disease. Even though these effector proteins could be targeted by the protein degradation systems in plant cells, they are able to perform their virulence functions, which suggests that they are relatively stable, even though the mechanisms leading to this stability remain poorly understood. In this work, we found that RipE1, an effector protein secreted by the bacterial wilt pathogen, *Ralstonia solanacearum*, has the potential to be ubiquitinated and degraded in plant cells. However, RipE1 hijacks plant kinases and undergoes phosphorylation of specific residues inside plant cells, and this counteracts its ubiquitination and promotes its stability. Moreover, RipE1 associates with plant ubiquitin proteases, which contribute to RipE1 deubiquitination and stabilization. Our study suggests that effector stability or degradation in plant cells constitute another molecular event subject to co-evolution

**Data availability statement:** All the data supporting the findings of this study are included in this article or as Supporting information.

**Funding:** This work was supported by the Research Fund for International Senior Scientists (RFIS-III) from the National Natural Science Foundation of China (grant 32350710194) to APM, the National Foreign Experts Program from the State Administration of Foreign Experts Affairs of China (grant H20240776) to APM, and the Shanghai Center for Plant Stress Biology (Center for Excellence in Molecular Plant Sciences, Chinese Academy of Sciences) to APM. The funders had no role in study design, data collection and analysis, decision to publish, or preparation of the manuscript.

**Competing interests:** The authors have declared that no competing interests exist.

between plants and pathogens, and that pathogen effectors hijack plant post-translational modification regulators in order to promote their own stability.

## Introduction

Plants and microbial pathogens have undergone a complex co-evolution process. As potential hosts without an adaptive immune system, plants have developed detection mechanisms to perceive pathogenic threats, activating immune responses and developmental rearrangements in order to hinder the development of disease [1]. Pathogens correspond to this with the development of virulence activities aimed at suppressing the activation of immune responses and manipulating other plant cellular functions in order to promote pathogen proliferation and, subsequently, the development of disease [2]. Studies on plant-pathogen interactions keep discovering new components of this co-evolution process, reflecting an unpredictable complexity.

One of the major virulence strategies of gram-negative bacterial pathogens is the injection of different effector proteins directly inside host cells using a type-III secretion system (T3SS). Those proteins, termed type-III effectors (T3Es) carry out numerous different virulence functions inside host plant cells, being essential for the development of disease [3-5]. However, as per the aforementioned co-evolution, plants have developed intracellular receptors carrying nucleotide-binding and leucine-rich repeat domains (NLRs) able to detect T3Es or their activities, by monitoring important plant proteins or cellular functions, perceiving perturbations in them, and subsequently activating immune responses [6]. Therefore, although T3Es are collectively essential for the development of disease, each one of them carries the potential of being perceived by the plant immune system in resistant host plants containing the appropriate NLRs. When a bacterial pathogen injects an effector that gets recognized by the plant immune system, pathogen evolution may drive the loss or modification of this effector [7], or the emergence of other effectors that may suppress this recognition or the subsequent downstream signalling [8,9]. In this situation, the recognized effector could exert its virulence activity without triggering immunity.

It is generally assumed that T3Es travel through the T3SS needle in an unfolded state [10]. Inside host cells, T3Es may associate with plant enzymes to undergo post-translational modifications that contribute to effector folding, subcellular localization, and/or function [11]. Besides achieving appropriate folding and functional protein conformations, it is worth considering the possibility that T3E proteins may be recognized by plant cells as non-self-proteins, and therefore be exposed to protein regulatory processes of plant cells, including their potential degradation [12,13]. Whether such regulatory processes constitute a threat for T3E proteins and how they escape from degradation is still poorly understood.

*Ralstonia solanacearum* is one of the most destructive bacterial plant pathogens worldwide. This soil-borne bacterium invades plants through the roots, reaching the vascular system, and subsequently colonizes the whole plant using xylem vessels [14]. *R. solanacearum* pathogenicity requires the activity of T3Es, and it is noteworthy that certain *R. solanacearum* strains secrete more than 70 different T3Es [15]. One of these T3Es, named RipE1, was originally validated as a protein injected into plant cells through the T3SS [16]; we subsequently found that RipE1 can be recognized by the plant immune system [17,18]. Despite RipE1 recognition, we have also found that *R. solanacearum* strains are able to secrete additional effectors to inhibit RipE1-triggered immunity, including RipAY [18,19], RipAC [20], and RipD [21]. In this work, we found that RipE1 undergoes phosphorylation and ubiquitination in plant cells. In the absence of phosphorylation, RipE1 ubiquitination is dramatically enhanced, and

the subsequent stability of RipE1 protein is severely reduced. Moreover, we found that RipE1 associates with plant ubiquitin proteases (ubiquitin carboxyl-terminal hydrolases, UCHs), which contribute to RipE1 de-ubiquitination and protein stability. Finally, we show that RipE1 phosphorylation and NbUCH-mediated deubiquitination are independent mechanisms that, together, lead to RipE1 stability in plant cells.

## Results

### RipE1 phosphorylation in plant cells contributes to protein stability

In order to understand better the biochemical properties of RipE1 in plant cells, we used Agrobacterium-mediated transient expression of *RipE1* fused to a green fluorescent protein (GFP) tag in leaves of the model *Solanaceae* plant *Nicotiana benthamiana*. We then performed affinity purification of RipE1-GFP followed by liquid chromatography-tandem mass spectrometry (LC-MS/MS) analysis. Among the resulting RipE1 peptides, we were able to identify five phosphorylated Serine (S)/Threonine (T) residues (Figs 1A, S1A **and** S1B). Given the close proximity of T53, S58, and S59, we generated a mutant RipE1 where these residues were replaced by Alanines (A) (RipE1³ᴬ) and a mutant where all five phosphorylated residues were mutated (RipE1⁵ᴬ). Upon expression in *N. benthamiana* leaves, both mutants showed lower protein accumulation compared to wild-type RipE1, and this phenomenon was particularly significant for the RipE1⁵ᴬ mutant (Fig 1B **and** 1C). All *RipE1* versions showed similar RNA accumulation (Fig 1D), indicating that the impaired protein accumulation is not caused by reduced gene expression, and suggesting that phosphorylation in these residues contributes to the stability of RipE1 protein in plant cells. Although wild-type RipE1 is stabilized by phosphorylation in plant cells, a phospho-mimic mutant in which all five phosphorylated residues were replaced by aspartic acid (D) (RipE1⁵ᴰ) showed a slightly stronger accumulation (Fig 1E-G), further supporting the notion that a negative charge in these residues (constitutive in this case) contributes to RipE1 stability.

In eukaryotic cells, ubiquitination of proteins may lead to their degradation through the 26S proteasome [12,13]. Interestingly, we found that the RipE1⁵ᴬ mutant is strongly ubiquitinated in plant cells (Fig 1H and 1I); the strong ubiquitination of both RipE1³ᴬ/⁵ᴬ mutants was particularly evident when protein loading was tuned to balance RipE1 accumulation (S2A-C Fig). Altogether, these results suggest that RipE1 phosphorylation in these residues is required to avoid RipE1 ubiquitination and maintain wild-type levels of RipE1 accumulation.

### RipE1 interacts with UBIQUITIN CARBOXYL-TERMINAL HYDROLASES in plant cells

Upon transient expression in *N. benthamiana* cells, RipE1-GFP is localized at the cell periphery, showing also a weak accumulation in the nucleus (S3A Fig). To identify RipE1-interacting proteins in plant cells, we searched for peptides present in RipE1 affinity-purified samples in the LC-MS/MS analysis mentioned above. Among peptides present in RipE1-GFP samples and absent in the GFP control, we found a remarkable accumulation of peptides corresponding to UBIQUITIN CARBOXYL-TERMINAL HYDROLASES (UCHs) (Fig 2A). The UCH family is not well defined in *N. benthamiana*, but the current version of the genome includes 13 genes encoding proteins annotated as UCHs and showing a percentage of identity higher than 50% when compared to the NbUCHs identified by LC-MS/MS and to their closest Arabidopsis homologs, AtUBP12 and AtUBP13 (Figs 2A, S3B and S1 Table).

Upon transient expression, a GFP-tagged version of NbUCH15 localizes at the cell periphery and the nucleus (S3C and S3D Fig), showing partial co-localization with RipE1 fused to a red fluorescent protein (RFP) tag (S3C and S3D Fig). We confirmed the physical

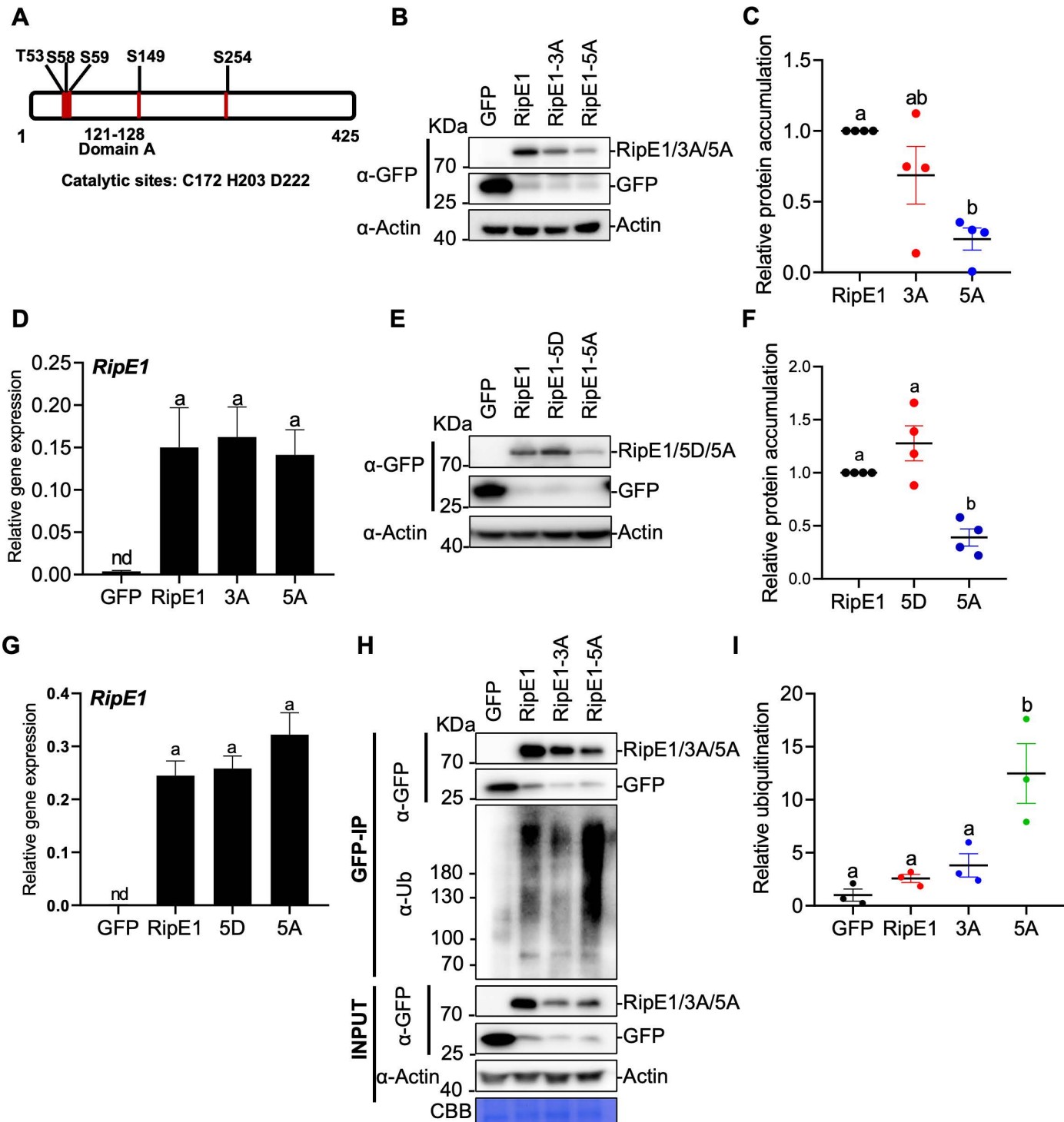

**Fig 1. RipE1 undergoes phosphorylation in *N. benthamiana*, which contributes to protein stability.** (A) Schematic representation of RipE1 protein, indicating the position of phosphorylation sites, domain A, and catalytic sites. (B) Western blot showing protein accumulation of GFP, RipE1-GFP, and RipE1 phosphodeficient mutants. *Agrobacterium* expressing GFP (as control), RipE1-GFP, 3A-GFP, or 5A-GFP (OD600 = 0.5) were infiltrated into the same leaf of *N. benthamiana*. Samples were taken at 30 hours post-infiltration (hpi), before the appearance of cell death. Blots were incubated with anti-Actin antibody to verify equal loading. This experiment was repeated 4 times, and the quantitation of the different repeats is shown in (C). (C) Quantification of the relative protein accumulation of the different repeats of the assay shown in (B), measured using Image J. RipE1 values were normalized using the respective actin values and represented as relative to RipE1 (WT) for each repeat. Values indicate mean ± SE (n = 4 biological replicates). Different letters indicate significant differences (one-way ANOVA, Tukey's test, p < 0.05). (D) Quantitative RT–PCR (qRT–PCR) to determine the expression of *RipE1* in (B). Expression values are relative to the expression of the housekeeping

gene *NbEF1a*. Values indicate mean ± SE (n = 9 biological replicates). Different letters indicate significant differences (one-way ANOVA, Tukey's test, p < 0.05). Composite data from 3 independent biological replicates. Nd: not detected. (E) Western blot showing protein accumulation of GFP, RipE1-GFP, 5D-GFP and 5A-GFP. This experiment was performed as in (B), repeated 4 times, and the quantitation of the different repeats is shown in (F). (F) Quantification of the relative protein accumulation of the different repeats of the assay shown in (E), measured and represented as in (C). (G) Quantitative RT–PCR (qRT–PCR) to determine the expression of *RipE1* in (E), performed and represented as in (D). (H) Immunoprecipitation assays to determine the ubiquitination status of RipE1 and phosphodeficient mutants. Samples were collected 30 hpi, before the appearance of cell death. Anti-GFP beads were used for immunoprecipitation. An anti-ubiquitin (P4D1) antibody was used to detect ubiquitinated proteins. Protein marker sizes are shown for reference. This experiment was repeated 3 times, and the quantitation of the different repeats is shown in (I). A similar assay where the sample loading was adjusted to show similar accumulation of all RipE1 variants for comparison of their ubiquitination is shown in S2A Fig. (I) Quantification of the relative protein ubiquitination of the different repeats of the assay shown in (H), measured using Image J. Ubiquitination values were normalized using the respective protein accumulation and represented as relative to the GFP control for each repeat. Values indicate mean ± SE (n = 3 biological replicates). Different letters indicate significant differences (one-way ANOVA, Tukey's test, p < 0.05).

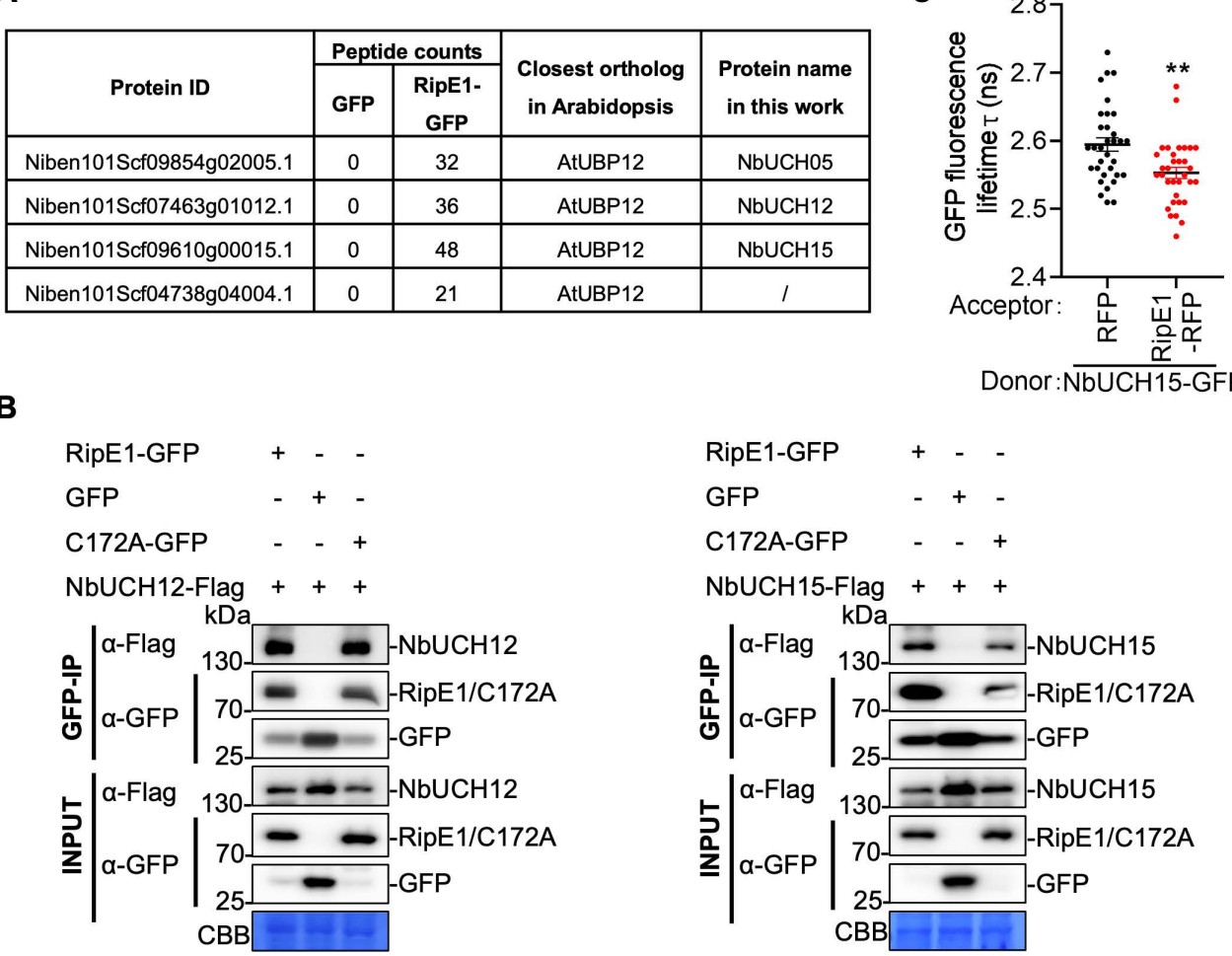

**Fig 2. RipE1 interacts with NbUCH12 and NbUCH15.** (A) Selected RipE1-GFP interactors identified by immunoprecipitation followed by LC-MS/MS. The table includes protein ID, peptide counts in GFP and RipE1-GFP samples, closest Arabidopsis orthologs, and the protein names used in this work. (B) Co-immunoprecipitation assays to determine interactions between RipE1 (WT or C172A mutant) and NbUCH12/15. *Agrobacterium* containing the indicated constructs were infiltrated in *N. benthamiana* leaves and samples were taken 2 days post-infiltration (dpi). Immunoblots were analyzed using anti-GFP and anti-Flag antibodies, and protein marker sizes are provided for reference. These experiments were repeated 3 times with similar results. (C) Interaction between RipE1-RFP and NbUCH15-GFP determined using FRET-FLIM upon transient co-expression in *N. benthamiana* leaves. Free RFP was used as a negative control. Fluorescence was visualized 30-36 hpi. Lines represent average values (n = 33), and error bars represent standard error. Asterisks indicate significant differences with the RFP control according to a Student's t-test (**p < 0.01). Composite data from 3 independent biological replicates.

association between RipE1 and NbUCH proteins by targeted Co-immunoprecipitation (CoIP) of RipE1 and two of the identified candidates, namely NbUCH12 and NbUCH15 (Fig 2B). A RipE1 mutant in the predicted catalytic cysteine (C172A), which does not trigger cell death [18] also associated with NbUCH12/15 (Fig 2B). We also measured Förster resonance energy transfer (FRET) between RipE1-RFP and NbUCH15-GFP using fluorescence life-time imaging (FLIM), and the results confirmed a direct interaction between RipE1 and NbUCH15 (Figs 2C and S3E).

## Reduced expression of *NbUCH* genes leads to the activation of immunity against *R. solanacearum*

Ubiquitin proteases encoded by *NbUCH* orthologs in Arabidopsis and *N. tabacum* have been previously described as negative regulators of immunity, since mutation or reduced expression of their corresponding genes enhance immune responses [22]. To perform a loss-of-function analysis of *NbUCH* genes, we generated constructs to silence them by RNA-interference (RNAi). RNAi constructs targeting either *NbUCH05*, *NbUCH12*, or *NbUCH15* led to a reciprocal reduced expression of all these three genes (S4A Fig), suggesting that either of these constructs causes a general silencing of *NbUCH* genes. The reduced expression of *NbUCH* genes led to the gradual emergence of tissue collapse (Fig 3A), which correlated with ion leakage indicative of cell death (Fig 3B). This also correlated with an enhanced expression of the defence-related marker gene *NbPR1* (Fig 3C) and an enhanced resistance of these tissues to *R. solanacearum* Y45 (Fig 3D), indicating that silencing of *NbUCH* genes leads to cell death associated to an activation of plant immunity.

The suppressor of the G2 allele of *skp1* (SGT1) is an essential component of the plant immune system, required for the induction of disease resistance mediated by many intracellular immune receptors containing nucleotide-binding and leucine-rich repeat domains (NLRs) [23,24]. Accordingly, virus-induced gene silencing (VIGS) of *SGT1* in *N. benthamiana* is commonly used to test for SGT1-dependent NLR-mediated cell death [25]. Interestingly, the cell death caused by the silencing of *NbUCH* genes was abolished by VIGS of *SGT1* (Fig 3E). This suggested two potential scenarios: (*i*) the previously described activity of UCHs as negative regulators of immunity requires SGT1, and/or (*ii*) a compromised accumulation or integrity of NbUCHs is perceived by an SGT1-dependent NLR, leading to the activation of immunity. The latter scenario prompted us to consider the following hypothesis: RipE1 could be targeting NbUCHs, and this targeting would lead to the activation of an NLR, and the subsequent activation of immune responses.

## The potential targeting of NbUCHs does not underlie RipE1-triggered immunity

We have recently found that RipE1 is recognized by the NLR NbPtr1 [17] (Kim et al., 2023). Accordingly, VIGS of *NbPtr1* abolishes the cell death and immune responses triggered by *RipE1* expression in plant cells (Fig 4A); [17]. To further analyze whether the observed activation of immunity upon *NbUCH* gene silencing could be related to RipE1-triggered immunity, we performed VIGS of *NbPtr1*. Interestingly, VIGS of *NbPtr1* did not affect the activation of immunity triggered by silencing *NbUCH* genes, as indicated by the development of cell death (Figs 4B and S4B) and the induction of *NbPR1* expression (Fig 4C), indicating that the activation of immunity triggered by RipE1 (NbPtr1-dependent) and that triggered by silencing *NbUCH* genes (Ptr1-independent) are based on different mechanisms. This suggests that the potential targeting of NbUCHs does not underlie RipE1-triggered immunity.

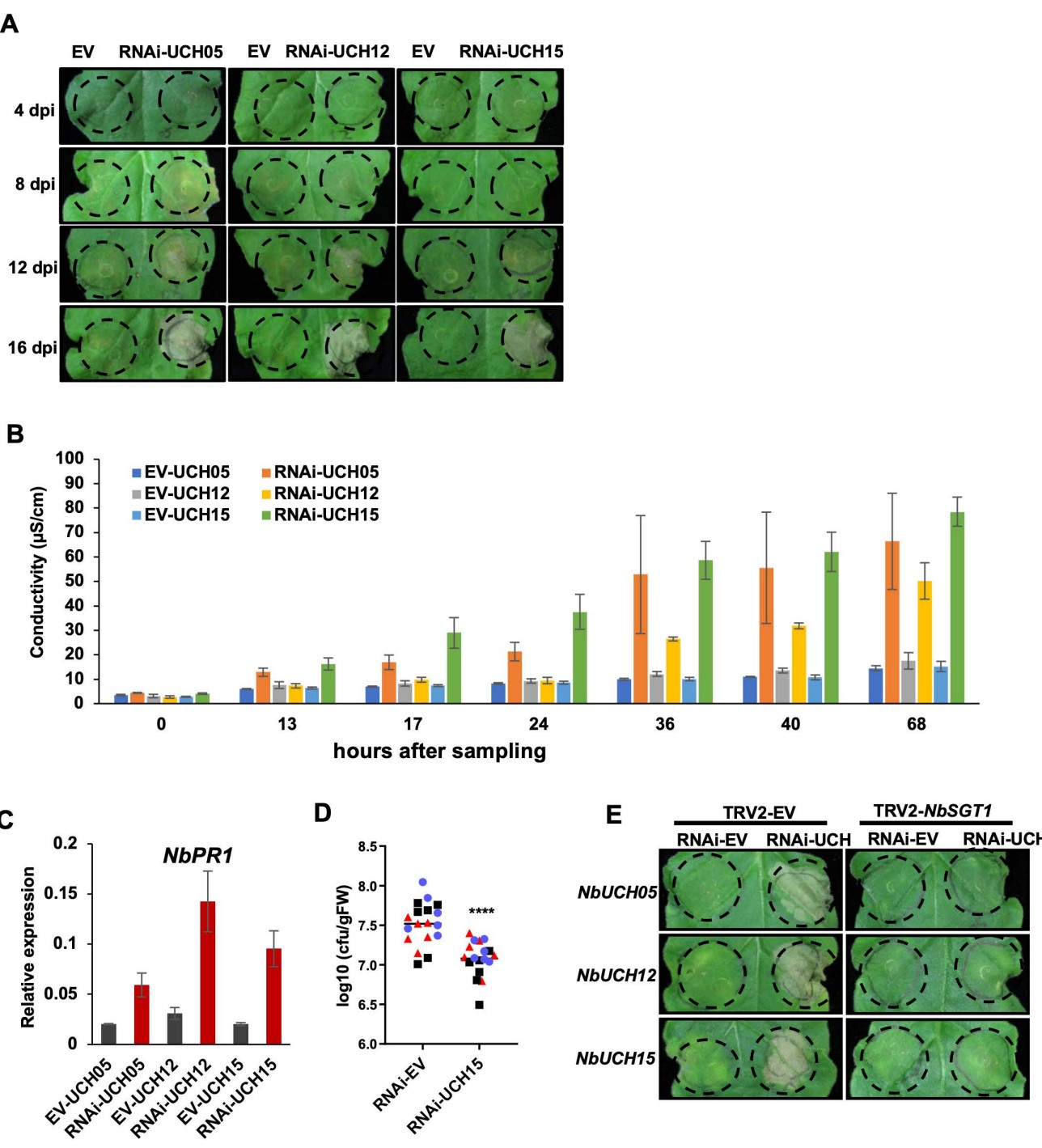

**Fig 3. Silencing *NbUCHs* triggers SGT1-dependent immunity in *N. benthamiana*.** *NbUCH* genes were silenced using RNAi in *N. benthamiana* leaves. An RNAi construct carrying the appropriate gene fragments (to silence each *NbUCH* gene) or an empty vector (as control) was expressed in the same leaf side-by-side using Agrobacterium (OD$_{600}$ = 0.5). (A) Tissue collapse, indicative of cell death, triggered by silencing *NbUCH* genes. Photographs were taken at the indicated days post-infiltration (dpi) using a CCD camera. The infiltrated areas are delimited using dotted lines. (B) Graph showing sample conductivity, indicating ion leakage from plant tissues caused by cell death. Leaf discs were collected 8 dpi , and ion leakage was measured at the indicated times after sampling. Values indicate mean ± SE (n = 3 biological replicates). (C) Quantitative RT-PCR to determine the expression of the defense-related gene *NbPR1*. Samples were taken 8 dpi . Expression values are relative to the expression of the housekeeping gene *NbEF1a*. Values indicate mean ± SE (n = 3 biological replicates). (D) Growth of *Ralstonia solanacearum* Y45 in *N. benthamiana*. *R. solanacearum* was inoculated into *N. benthamiana* leaves after silencing *NbUCH15* for 8 days, before the appearance of cell death. Leaf discs were collected 2 dpi for

bacterial quantification. Values indicate mean ± SEM (n = 18 biological replicates from 3 independent repeats; each color corresponds to values from an independent replicate. Asterisks indicate significant differences compared to control according to a Student's t test (**** p < 0.0001). (E) RNAi constructs for silencing *NbUCHs* or an empty vector (as control) were expressed in *N. benthamiana* undergoing VIGS of *NbSGT1* or VIGS using an empty vector ( EV ) construct (as control). Tissue collapse, indicative of cell death, was recorded 16 dpi using a CCD camera. The infiltrated areas are delimited using dotted lines. Each experiment was repeat at least three times with similar results.

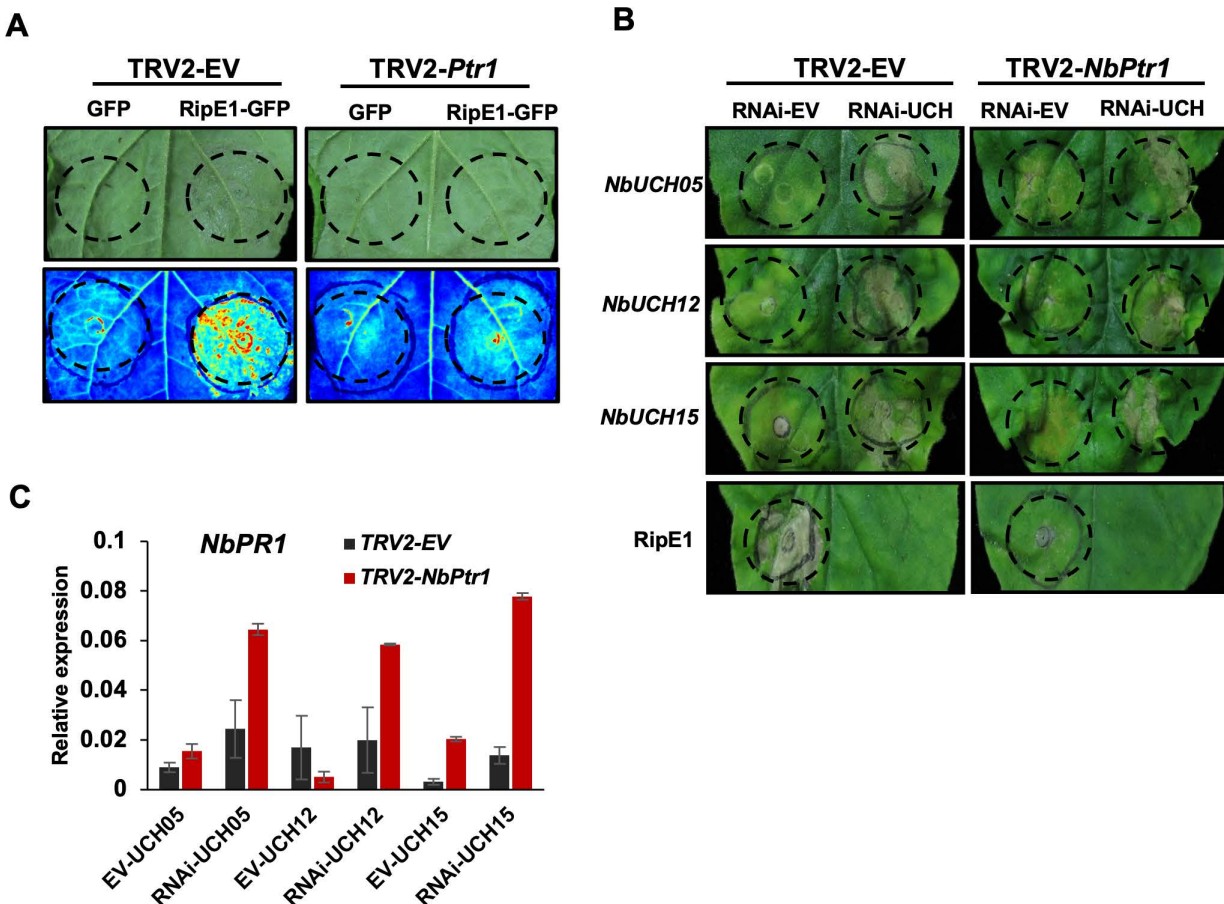

**Fig 4. Immunity triggered by silencing *NbUCH* does not require *NbPtr1*.** Agrobacterium expressing an empty vector or a construct to induce virus-induced gene silencing (VIGS) of *NbPtr1* were infiltrated *in N.* benthamiana leaves. (A) Nine days after infiltration to induce VIGS of *NbPtr1*, Agrobacterium expressing RipE1 or a GFP control were infiltrated into *N. benthamiana* leaves. Tissue collapse, indicative of cell death, triggered by RipE1 expression was recorded 2 dpi. The infiltrated areas are delimited using dotted lines. Upper photographs were taken using a CCD camera from the adaxial side of the leaves, and lower pictures were captured using a UV camera from the abaxial side of the leaves and were flipped horizontally for representation. (B and C) Nine days after infiltration to induce VIGS of *NbPtr1*, RNAi constructs for silencing *NbUCHs* or an empty vector (as control) were expressed in *N. benthamiana* leaves. (B) Tissue collapse, indicative of cell death, was recorded 12 dpi using a CCD camera. The infiltrated areas are delimited using dotted lines. Tissue collapse triggered by RipE1 (4 dpi) was used as control. (C) Quantitative RT-PCR to determine the expression of the defense-related gene *NbPR1*. Samples were taken 8 dpi. Expression values are relative to the expression of the housekeeping gene *NbEF1a*. Values indicate mean ±SE (n = 3 biological replicates).

## NbUCHs contribute to RipE1 deubiquitination and protein stability in plant cells

After considering unlikely that NbUCHs are virulence targets responsible for RipE1-triggered immunity, we considered other potential functional associations between RipE1 and NbUCHs. Interestingly, in tissues undergoing RNAi-mediated silencing of *NbUCH* genes, we

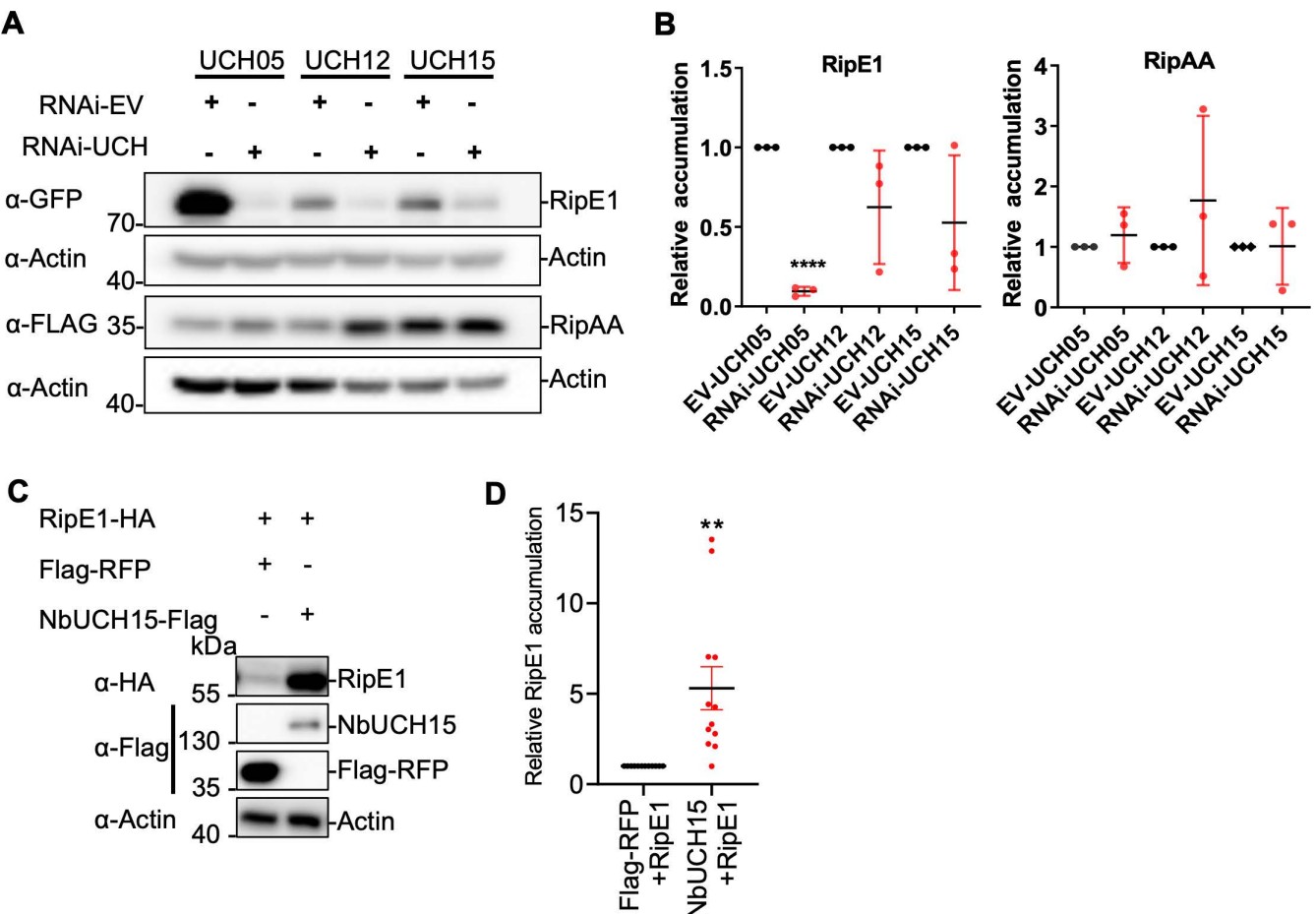

**Fig 5. NbUCH positive regulates RipE1 accumulation in *N. benthamiana*.** (A and B) *NbUCH* genes were silenced using RNAi in *N. benthamiana* leaves. An RNAi construct carrying the appropriate gene fragments (to silence each *NbUCH* gene) or an empty vector (as control) was expressed in the same leaf side-by-side using Agrobacterium ($OD_{600}$=0.5). One day later, RipE1-GFP or RipAA-FLAG (as control) were expressed in same leaf using Agrobacterium ($OD_{600}$=0.1). (A) Western blot showing the accumulation of RipE1-GFP and RipAA-FLAG protein accumulation. Blots were incubated with anti-Actin antibody to verify equal loading. Protein marker sizes are shown for reference. This experiment was repeated 3 times, and the quantitation of the different repeats is shown in (B). (B) Quantification of the relative protein accumulation of the different repeats of the assay shown in (A), measured using Image J. RipE1 or RipAA values were normalized using the respective actin values and represented as relative to their respective empty vector control for each repeat. Values indicate mean ± SE (n = 3 biological replicates). Asterisks indicate significant differences compared to each control according to a Student's t test (* $p < 0.05$, ** $p < 0.01$). (C) Western blot to determine RipE1 protein accumulation after expression of NbUCH15 or Flag-RFP (as control). Agrobacterium expressing RipE1 was infiltrated 24 hours after expression of Flag-RFP (as control) or NbUCH15-Flag side-by-side in the same *N. benthamiana* leaf. Blots were incubated with anti-Actin antibody to verify equal loading. Protein marker sizes are shown for reference. This experiment was repeated 12 times, and the quantitation of the different repeats is shown in (D). (D) Quantification of the relative protein accumulation of the different repeats of the assay shown in (C), measured using Image J. RipE1 values were normalized using the respective actin values and represented as relative to the control expressing Flag-RFP for each repeat. Values indicate mean ± SE (n = 12 biological replicates). Asterisks indicate significant differences compared to each control according to a Student's t test (** $p < 0.01$).

observed a reduced accumulation of RipE1 protein upon Agrobacterium-mediated transient expression (Fig 5A). *NbUCH* silencing did not significantly reduce the accumulation of a different *R. solanacearum* T3E, RipAA, in the same conditions (Fig 5A **and** 5B) or the accumulation of *RipE1* RNA (S5 Fig). These results suggest that *NbUCH* silencing has a specific impact over the accumulation of RipE1 protein. To confirm this hypothesis, we overexpressed *NbUCH15* and subsequently expressed *RipE1*. The overexpression of *NbUCH15* led to a significantly enhanced accumulation of RipE1-GFP (Fig 5C and 5D), indicating that NbUCH15 promotes RipE1 accumulation.

Given the direct correlation between *NbUCH* expression and RipE1 protein accumulation, and our previous observation that RipE1 undergoes ubiquitination in plant cells (Fig 1), we considered the possibility that NbUCHs mediate the de-ubiquitination of RipE1, therefore promoting its stability. Indeed, we observed much stronger RipE1 ubiquitination upon silencing of *NbUCH15* (Figs 6A, 6B and S6) and a significant inhibition of RipE1 ubiquitination upon *NbUCH15* overexpression (Fig 6C and 6D). Given that the alteration of *NbUCH* gene expression leads to different accumulation of RipE1 protein, we tuned the loading of the samples in order to compare the ubiquitination of similar levels of RipE1, as indicated in the Fig panels (Fig 6A-D). These results suggest that NbUCHs contribute to RipE1 stability by reducing its ubiquitination.

### RipE1 phosphorylation and NbUCH-mediated deubiquitination are independent mechanisms leading to RipE1 stability

Given that both RipE1 phosphorylation and its association with NbUCHs contribute to RipE1 stability, we tested whether RipE1 phosphorylation is somehow required for the association with NbUCHs and subsequent deubiquitination. In order to test the interaction between NbUCH15 and RipE1/RipE1[5A], we expressed RipE1/RipE1[5A] from an inducible promoter and collected protein samples at an early time-point, when RipE1[5A] accumulation is still comparable to wild-type RipE1. In these conditions, the RipE1[5A] mutant showed a similar association with NbUCH15 in CoIP assays (Fig 7A), indicating that RipE1 phosphorylation in these residues is not required for association with NbUCH15.

Overexpression of NbUCH15 was also able to rescue the low protein accumulation observed for RipE1[5A] (Fig 7B and 7C), likely as a consequence of a strong inhibition in RipE1[5A] ubiquitination mediated by NbUCH15 (Fig 7D and 7E); protein loading was tuned to analyze similar amounts of RipE1[5A]). Altogether, these results suggest that RipE1 phosphorylation and NbUCH-mediated deubiquitination are independent mechanisms that, together, lead to RipE1 stability in plant cells.

### Discussion

Unlike fungal, oomycete, or viral effector proteins, which are synthesized in eukaryotic cells, bacterial T3Es travel unfolded through the T3SS and, as prokaryotic polypeptides, need to achieve active conformations and the appropriate subcellular localization for their respective functions in host cells [10,11]. Given that eukaryotic cells have mechanisms to recognize and deal with unnatural proteins [26], it is unclear how these T3E polypeptides, as non-plant proteins, survive the protein degradation machinery in plant cells. In animal pathogenic bacteria, such as *Salmonella*, certain T3Es have been shown to regulate their function sequentially through their proteasome-mediated degradation [27], although such mechanisms have not yet been described for plant pathogenic bacteria. In the case of T3Es that target the host protein degradation machinery [28,29], their actual virulence activity may protect themselves against host-mediated degradation. Such is the case of the *Xanthomonas campestris* T3E XopL, which has been recently shown to be degraded by autophagy [30]; however, XopL inhibits autophagy in plant cells, promoting pathogen infection and its own protein stability [30]. In this work, we found that RipE1 is ubiquitinated in plant cells, and this ubiquitination promotes effector instability or degradation. This may constitute a defence mechanism against potentially threatening non-self proteins in plant cells. Despite such ubiquitination, we found that RipE1 stability in plant cells is promoted by phosphorylation and deubiquitination mediated by plant deubiquitinating enzymes. Unlike T3Es like XopL, which promote their stability by inhibiting plant defence mechanisms (*e.g.,* autophagy), RipE1 seems to hijack the post-translational modification (PTM) machinery in host cells to promote its own stability.

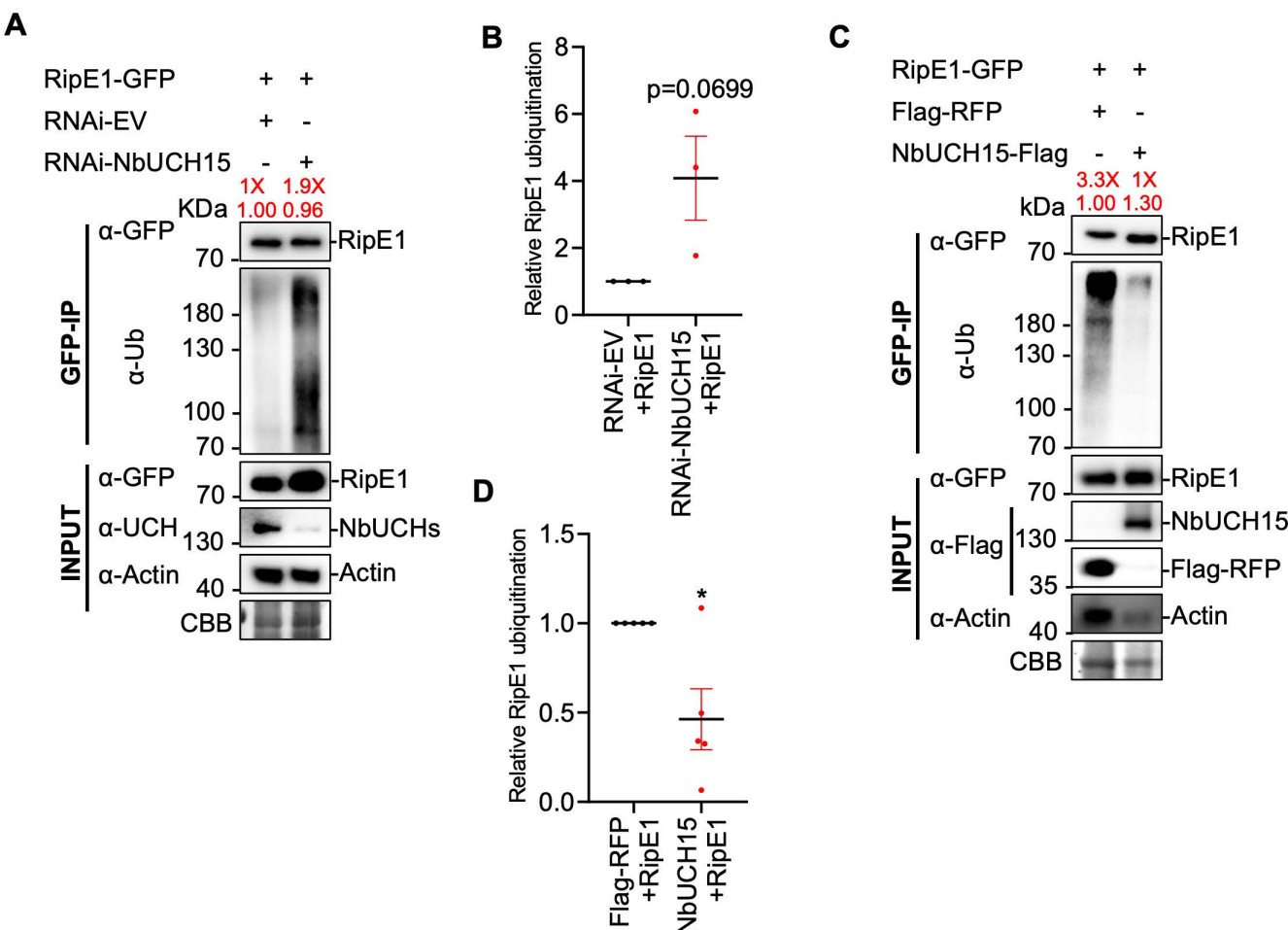

**Fig 6. NbUCH15 promotes RipE1 stability by deubiquitination.** (A and C) Immunoprecipitation assays to determine the ubiquitination status of RipE1. (A) Agrobacterium expressing RipE1-GFP was infiltrated 1 day after RNAi-mediated silencing of *NbUCH15*. An empty vector was used as RNAi negative control. Samples were collected 30 hpi, before the appearance of cell death. Given that *NbUCH15* silencing compromises RipE1 protein accumulation, different volumes of the protein samples were loaded to show a comparable RipE1-GFP protein accumulation between different lanes after immunoprecipitation, allowing the detection of ubiquitination in the same amount of RipE1 protein. The relative loading volumes and protein abundance are indicated above lanes. A similar assay showing the loading of the same sample volumes (with different RipE1 accumulation) is shown in S6A Fig. Anti-GFP beads were used for immunoprecipitation. An anti-ubiquitin (P4D1) antibody was used to detect ubiquitinated proteins. The accumulation of native NbUCH proteins was detected using a custom anti-NbUCH antibody and the same volume of each sample. Protein marker sizes are shown for reference. This experiment was repeated 3 times, and the quantitation of the different repeats is shown in (B). (B) Quantification of the relative RipE1 ubiquitination of the different repeats of the assay shown in (A), measured using Image J. Ubiquitination values were normalized using the respective protein accumulation and represented as relative to the empty vector control for each repeat. Values indicate mean ± SE (n = 3 biological replicates). The p value compared to the control value using a Student's t test is shown. (C) Agrobacterium expressing *RipE1-GFP* was infiltrated 1 day after expression of *NbUCH15-Flag* or *Flag-RFP* (as control). Samples were collected 30 hpi, before the appearance of cell death. Given that *NbUCH15* overexpression enhances RipE1 protein accumulation, different volumes of the protein samples were loaded to show a comparable RipE1-GFP protein accumulation between different lanes after immunoprecipitation, allowing the detection of ubiquitination in the same amount of RipE1 protein. The relative loading volumes and protein abundance are indicated above lanes. Anti-GFP beads were used for immunoprecipitation. An anti-ubiquitin (P4D1) antibody was used to detect ubiquitinated proteins. Protein marker sizes are shown for reference. This experiment was repeated 5 times, and the quantitation of the different repeats is shown in (D). (D) Quantification of the relative RipE1 ubiquitination of the different repeats of the assay shown in (C), measured using Image J. Ubiquitination values were normalized using the respective protein accumulation and represented as relative to the control expressing Flag-RFP for each repeat. Values indicate mean ± SE (n = 5 biological replicates). Asterisks indicate significant differences compared to the control according to a Student's t test (* p < 0.05).

Several bacterial effectors have been found to undergo PTMs and to associate with eukaryotic factors in host cells, which contribute to their virulence activity by mediating their biochemical activation and/or subcellular localization [11]. Phosphorylation in plant cells has been reported

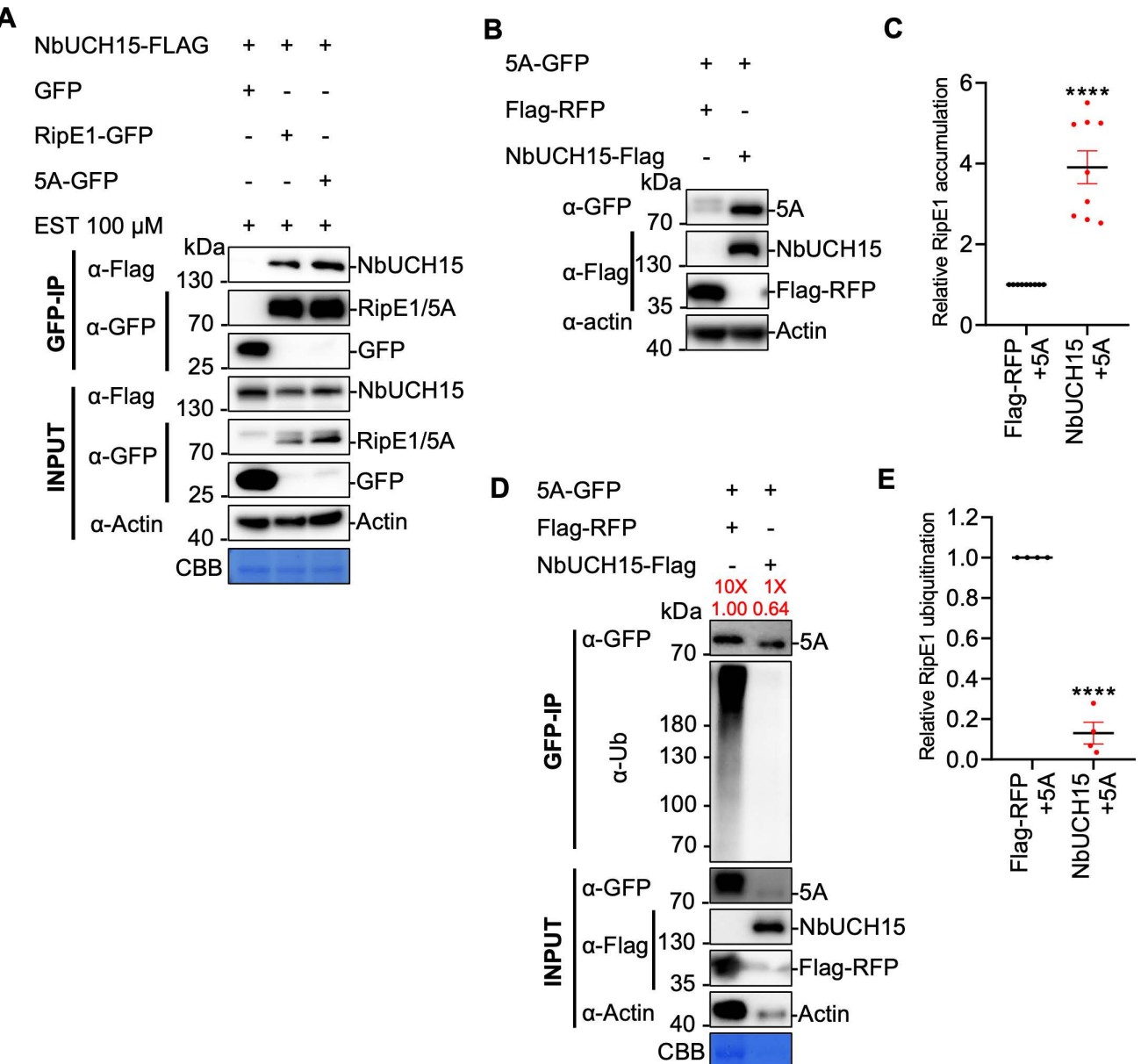

**Fig 7. RipE1 phosphorylation and NbUCH-mediated deubiquitination are independent mechanisms leading to RipE1 stability.** (A) Co-immunoprecipitation assay to analyse the interaction between RipE1-5A and NbUCH15. *Agrobacterium* containing the indicated constructs were infiltrated in *N. benthamiana* leaves. In order to avoid stability issues with the RipE1-5A mutant, an estradiol (EST)-inducible vector was used, and samples were treated with 100 μM EST for 2.5 hours before being harvested at 2.5 dpi. Immunoblots were analyzed using anti-GFP and anti-Flag antibodies, and protein marker sizes are provided for reference. These experiments were repeated 3 times with similar results. (B) Western blot to determine the accumulation of the RipE1-5A mutant after expression of NbUCH15 or Flag-RFP (as control). Agrobacterium expressing the RipE1 versions were infiltrated 24 hours after expression of Flag-RFP (as control) or NbUCH15-Flag side-by-side in the same *N. benthamiana* leaf. Blots were incubated with anti-Actin antibody to verify equal loading. Protein marker sizes are shown for reference. This experiment was repeated 9 times, and the quantitation of the different repeats is shown in (C). (C) Quantification of the relative protein accumulation of the different repeats of the assay shown in (B), measured using Image J. RipE1 values were normalized using the respective actin values and represented as relative to the control expressing Flag-RFP for each repeat. Values indicate mean ± SE (n = 9 biological replicates). Asterisks indicate significant differences compared to the control according to a Student's t test (**** $p < 0.0001$). (D) Agrobacterium expressing *RipE1-5A-GFP* was infiltrated 1 day after expression of *NbUCH15-Flag* or *Flag-RFP* (as control). Samples were collected 30 hpi, before the appearance of cell death. Given that *NbUCH15* overexpression enhances RipE1 protein accumulation, different volumes of the protein samples were loaded to show a comparable RipE1-GFP protein accumulation between different lanes after immunoprecipitation, allowing the detection of ubiquitination in the same amount of RipE1 protein. The relative loading volumes and protein abundance are indicated above lanes. Anti-GFP beads were used for immunoprecipitation. An anti-ubiquitin (P4D1) antibody was used to detect ubiquitinated proteins. Protein marker sizes are shown for reference. This experiment was repeated 4 times, and the quantitation of the different

repeats is shown in (E). (E) Quantification of the relative RipE1 ubiquitination of the different repeats of the assay shown in (D), measured using Image J. Ubiquitination values were normalized using the respective protein accumulation and represented as relative to the control expressing Flag-RFP for each repeat. Values indicate mean ± SE (n = 4 biological replicates). Asterisks indicate significant differences compared to the control according to a Student's t test (**** $p < 0.0001$).

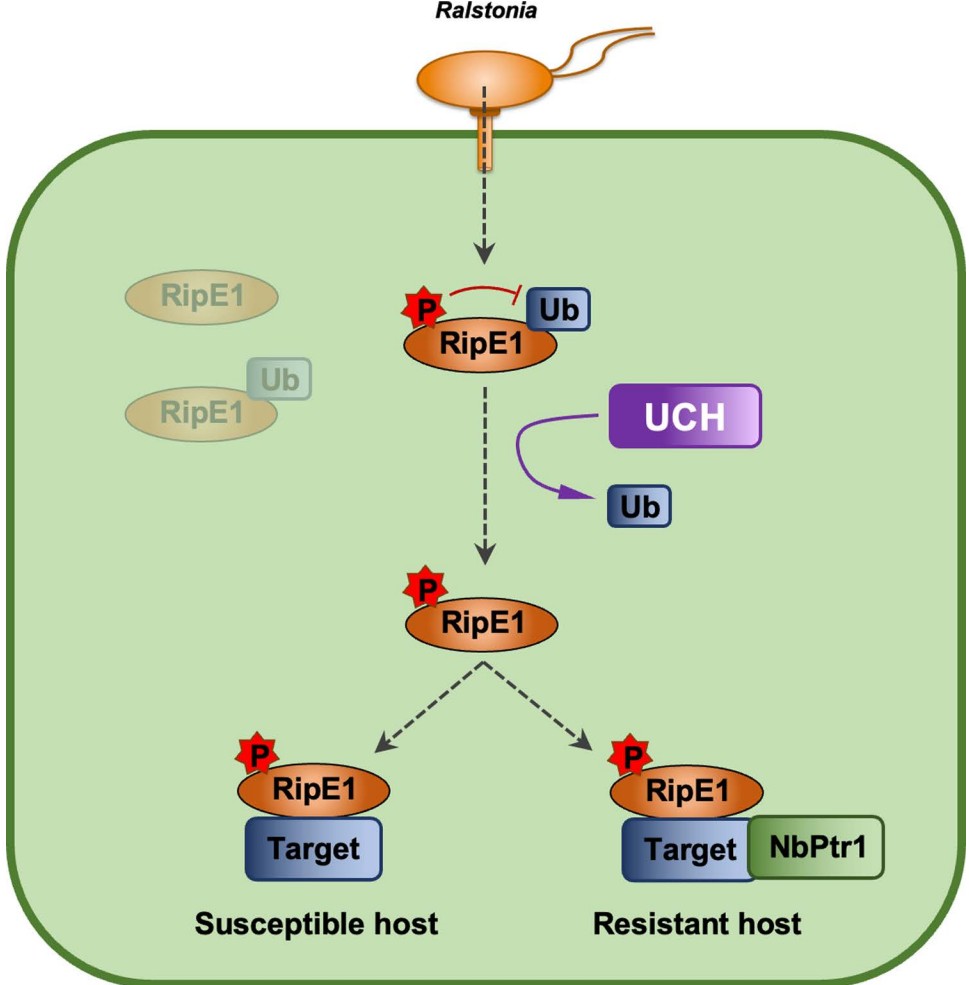

**Fig 8. Simplified diagram showing a schematical model of RipE1 protein stability and activity in plant cells.** Upon injection inside plant cells, RipE1 is subjected to ubiquitination and subsequent degradation. RipE1 phosphorylation and UCH-mediated deubiquitination contribute to RipE1 stability. In susceptible hosts, phosphorylated stable RipE1 exerts its virulence activity through the association with virulence target(s). In resistant hosts, RipE1 activity is perceived by the presence of NbPtr1, leading to the activation of immune responses and disease resistance.

for several T3Es [11]; among *R. solanacearum* T3Es, RipAY associates with thioredoxins and undergoes phosphorylation in plant cells in order to activate its GGCT activity [19,31,32]. In the case of RipE1, phosphorylation seems to prevent RipE1 ubiquitination and promote RipE1 stability, which could be a way for the effector to adapt to the threat of degradation by the plant protein degradation machinery (Fig 8). Interestingly, the identified phosphorylation sites are conserved among RipE1 sequences from different *R. solanacearum* strains (S7 Fig), suggesting that the phosphorylation of these residues may be an important mechanism contributing to RipE1 stability and subsequent function. We hypothesize that RipE1 phosphorylation may

directly counteract its ubiquitination, but it is also possible that phosphorylation is required for the appropriate folding of RipE1, and that the mutation of the phosphorylated residues to the non-phosphorylable Alanine causes a severe misfolding that exacerbates the ubiquitination and degradation of RipE1. In addition to this phosphorylation-mediated mechanism to promote protein stability, we also found that RipE1 associates with specific plant ubiquitin proteases from the UCH family. Plant UCHs seem to deubiquitinate RipE1 and promote effector stability; this also constitutes a novel adaptation mechanism for the pathogen, hijacking plant ubiquitin proteases to counteract ubiquitination in plant cells and subsequent degradation in order to promote effector virulence activities (Fig 8). However, we should not completely rule out an alternative scenario, in which RipE1 may have evolved to transiently perform its virulence function and be subsequently degraded. In this case, plant species harbouring an NLR recognizing RipE1 activity may promote effector stability using phosphorylation and/or UCH-mediated deubiquitination, as a mechanism of plant cells to enhance RipE1 detection and the subsequent activation of immunity. However, such mechanism would only be reasonable in the context of RipE1-triggered immunity, which, to date, has only been described in *N. benthamiana*.

RipE1 is conserved in most *R. pseudosolanacearum* strains (corresponding to the phylotype I of the *R. solanacearum* species complex) sequenced to date [15,17]. Once RipE1 is secreted inside plant cells and overcomes the threat of degradation, as characterized in this work, RipE1 activities could be detected in resistant host plants containing the NLR Ptr1 [17], which would lead to the activation of disease resistance [17,18]. Instead of losing RipE1 as a mean of pathogenic adaptation to resistant host plants, these strains contain other effectors with the potential to mask or suppress RipE1-triggered immunity [18,20,21]. The strong conservation of RipE1 among *R. pseudosolanacearum* strains and the complex evolution undergone to keep RipE1 stable and counteract its potential detection suggest that RipE1 may play an important role in pathogen virulence. Further work will be required to characterize RipE1 contribution to *R. pseudosolanacearum* infection.

RipE1-triggered immunity in *N. benthamiana* requires its predicted cysteine-protease catalytic activity [18] and the presence of NbPtr1 [17]. After finding that silencing NbUCH triggers immune-associated HR, we first hypothesized that NbUCHs may be targets of RipE1, and this activity could be monitored by NbPtr1 to activate immune responses. However, two observations suggest that NbUCHs are not virulence targets of RipE1 that are guarded by NbPtr1: first, a RipE1-C172A mutant, which loses its catalytic activity and is not recognized by NbPtr1, still associates with UCHs (Fig 2); second, the HR triggered by the lack of UCHs is not abolished by silencing NbPtr1 (Fig 4). This prompted us to explore a different scenario to explain the association between RipE1 and NbUCHs. Our subsequent data suggests that NbUCHs associate with RipE1, promoting RipE1 deubiquitination and RipE1 stability; in this sense, NbUCHs could act as "helpers" for the effector to contribute to its stability and activity (Fig 8). UCHs are essential proteins for plant cells. In particular, the Arabidopsis homologs of the NbUCHs characterized in this work, AtUBP12/AtUBP13, constitute an important regulatory node in different signaling pathways [33]. Among other functions, AtUBP12/AtUBP13 have been shown to belong to the polycomb group protein system to regulate gene silencing [34], and to contribute to the stability of MYC2 to activate jasmonate-dependent responses [35]. Besides their initial characterization as negative regulators of immunity [22], it has been recently shown that AtUBP12/AtUBP13 mediate the deubiquitination of the salicylic acid receptor NPR3 to suppresses plant immunity, playing an important role in the negative regulation of immune activation [36]. Given the essential activities carried out by the UBP/UCH-family proteins in plant cells, RipE1 hijacks an important regulatory node, which cannot be "shut down" by plant cells, in order to prevent its own degradation.

Finally, we set out to determine whether RipE1 phosphorylation may directly counteract ubiquitination or may otherwise contribute to the interaction with NbUCHs. Our results were clear to show that RipE1 phosphorylation in the reported 5 residues is not required for interaction with NbUCH15 (Fig 7). Moreover, overexpression of NbUCH15 was also able to rescue the low protein accumulation observed for RipE1[5A], likely as a consequence of a strong inhibition in RipE1[5A] ubiquitination mediated by NbUCH15, suggesting that phosphorylation and NbUCH15-mediated deubiquitination are independent processes and may have an additive effect contributing to RipE1 stability in plant cells. It is worth noting that, similar to the case of RipE1, we have recently found that another *R. solanacearum* T3E, RipBM, also undergoes phosphorylation and associates with plant 14-3-3 proteins to prevent its degradation. These results suggest that phosphorylation and association with important plant proteins may be a common strategy among T3Es to counteract degradation and promote their stability in plant cells.

## Materials and Methods

### Plant materials

*N. benthamiana* plants were grown at 23°C in a growth room under 16-h light/8-h dark photoperiod with a light-intensity of 130 mE $m^{-2}s^{-1}$. Each plant was grown in one pot on soil with 1:1 mix of potting soil and vermiculite. After *R. solanacearum* inoculation, plants were moved to a 27°C growth chamber with 75% humidity under a 14-h light/10-h dark photoperiod.

### Bacterial strains

*Agrobacterium tumefaciens* GV3101 carrying different constructs was grown on solid Luria-Bertani (LB) medium plates with the appropriate antibiotics. The concentration of each antibiotic was 25 µg $mL^{-1}$ rifampicin, 50 µg $mL^{-1}$ gentamicin, 50 µg $mL^{-1}$ spectinomycin, and 50 µg $mL^{-1}$ kanamycin.

*R. solanacearum* Y45 was grown on solid Bacto-agar and Glucose (BG) medium for 2 days at 28°C. Then bacteria were cultivated overnight after inoculation in liquid BG medium.

### Generation of plasmid constructs

The generation of expression plasmids for *ripE1* from *R. solanacearum* GMI1000 (Rsc3369) was previously described [18]. The *ripE1* fragment in pDONR207 was used as template to generate *ripE1* phosphosite mutants by site-directed mutagenesis using the QuickChange Lightning Site-Directed Mutagenesis Kit (Life technologies, USA) following the manufacturer's instructions and the primers indicated in S2 Table. The triple 3A mutant fragment was cloned into pENTR-D-TOPO (Thermo Fisher Scientific, MA, USA). The 3A mutant was used as a template to generate the 5A mutant. RipE1 and 5A with addition of an eGFP fragment from pGWB505 were inserted into pER8 vector using one-step cloning recombination (Vazyme, China). The pDONR207-RipE1 was used as template to generate *ripE1* phospho-mimic mutants by site-directed mutagenesis using fast site-directed mutagenesis kit (Vazyme, China) following the manufacturer's instructions. The pDONR207-RipE1-5D was sub-cloned into pGWB505 via LR reaction, resulting in pGWB505-RipE1-5D.

*NbUCH12* (Niben101Scf07463g01012.1) and *NbUCH15* (Niben101Scf09610g00015.1) CDS fragments were cloned into pEASY-Blunt. *NbUCH15* or *NbUCH12* full length CDS fragments were cloned into pCAMBIA1300-FLAG vector via digestion and ligation or one-step cloning recombination (Vazyme, China).

*NbUCH* fragments to target specific genes using an RNAi approach was designed using the Solgenomics VIGS online tool (https://vigs.solgenomics.net/). The pK7GWIWG2 II-RedRoot expression vector [37] was used for transient silencing. Each *NbUCH* specific fragment was amplified using the primers described in S2 Table and cloned into pENTR-D-TOPO, and then sub cloned into pK7GWIWG2-II via LR recombination.

All the primer sequences are shown in S2 Table.

## Transient expression in *N. benthamiana*

Agrobacteria carrying the indicated constructs were infiltrated into 4-5-week-old *N. benthamiana*. The bacterial density used was $OD_{600}$ 0.2-0.5 for confocal microscopy, protein accumulation, and phenotypic assays, $OD_{600}$ 1.0 was used for VIGS assays, and $OD_{600}$ 0.25 or 0.5 was used for RNAi-mediated local silencing. Bacteria were suspended in infiltration buffer (10 mM $MgCl_2$, 10 mM MES pH 5.6, and 150 μM acetosyringone), and then infiltrated into plant leaves using a 1 mL needless syringe.

## Confocal microscopy

Confocal microscopy was performed as previously described [38]. Leaf epidermal cells of *N. benthamiana* were examined using a Leica TCS SP8 confocal microscope (Leica, Germany). The settings for excitation and emission were 488 nm (ex) and 500-550 nm (em) for GFP, and 561 nm (ex) and 580–630 nm (em) for RFP. Sequential scanning was used to avoid interference between the GFP and RFP channels.

## Conductivity measurements

Cell death in plant leaves was quantified as previously described [18,20] by measuring the electrolyte leakage using a conductivity meter (ThermoFisher, USA) or observing the autofluorescence using the BioRad Gel Imager (Bio-Rad, USA). Briefly, one day after Agrobacterium infiltration in *N. benthamiana*, one 13 mm leaf disk was immersed in 4 mL of distilled water for 1 h with gentle shaking and then transferred to a 6-well culture plate containing 4 mL distilled water in each well. The ion conductivity was then measured at different time intervals. Autofluorescence in intact *N. benthamiana* leaves was measured at 2.5 dpi.

## Protein extraction and Western blots

Protein extraction and western blots were performed as previously described [18,20]. Briefly, plant tissues were frozen and ground in liquid nitrogen and ground using a Tissue Lyser (QIAGEN, Hilden, Nordrhein-Westfalen, Germany) with a frequency of 25 s$^{-1}$ for 1 min. Ground tissues were then homogenized in protein extraction buffer (100 mM Tris pH 8, 150 mM NaCl, 10% glycerol, 5 mM EDTA, 2 mM DTT, 1% (v/v) protease inhibitor cocktail, 2 mM PMSF, 1% (v/v) NP40, 10 mM sodium molybdate, 10 mM sodium fluoride, 2 mM sodium orthovanadate). The resulting protein samples were incubated at 70°C for 10 minutes in SDS loading buffer and loaded in SDS-PAGE acrylamide gels for western blot analysis. Immunoblots were analyzed using the antibodies indicated in the figures: anti-GFP (Abicode, M0802-3a), anti-luciferase (Sigma, L0159), anti-actin (Agrisera, AS13 2640), anti-ubiquitination(P4D1) (Santa Cruz Biotechnology sc-8017 HRP), anti-FLAG (Abmart, M20008), anti-RFP (Chromotek 5F8), antibodies. The custom antibody against NbUCH15, was generated by Abclonal. Protein signals were quantified using the Image J software.

## Chemical treatments

MG132 (Sangon Biotech, China) powder was dissolved in DMSO to a final concentration at 10 mM. 50 μM MG132 diluted in water was infiltrated into plants 4 hours before collecting samples. In all assays to detect ubiquitination, samples were collected after 50 μM MG132 treatment for 4 hours.

## Co-immunoprecipitation

Co-immunoprecipitation assays were performed as previously described [20]. Briefly, 500 mg of ground *N. benthamiana* leaves were resuspended in 1 mL protein extraction buffer as indicated above. Supernatants were filtered through micro bio-spin chromatography columns. The filtered extracts were incubated with 15 μL GFP-trap agarose beads (ChromoTek, Germany) at 4°C for 1 hour, followed by 4 washes with wash buffer (100 mM Tris-HCl pH 8, 150 mM NaCl, 10% glycerol, 2 mM DTT, 1% (v/v) protease inhibitor cocktail, 0.5% (v/v) NP40, 10 mM sodium molybdate, 10 mM sodium fluoride, 2 mM sodium orthovanadate). To detect ubiquitination, 50 μM MG132 and 10 mM NEM were added to the protein extraction buffer and 20 μM MG132 and 2.5 mM NEM were added to the wash buffer. The resulting protein samples were incubated at 70°C for 10 minutes in SDS loading buffer and loaded in SDS-PAGE acrylamide gels for western blot analysis.

## Large-scale immunoprecipitation and LC-MS/MS analysis

Large-scale immunoprecipitation assays for LC-MS/MS analysis were performed as previously described [20]. Briefly, 5 g of ground *N. benthamiana* leaves and 50 μL GFP-trap agarose beads were used following the co-immunoprecipitation procedure described above. Two more washes with wash buffer without NP40 were performed before LC-MS/MS analysis.

## RNA extraction and quantification RT-PCR

Plant tissues were collected into 2 mL tubes with one metal bead, frozen in liquid nitrogen and then ground using a Tissue Lyser (QIAGEN, Hilden, Nordrhein-Westfalen, Germany) with a frequency of $25 \, s^{-1}$ for 1 min. Total RNA was extracted using E.Z.N.A. Plant RNA kit (Biotek, China) following the the manufacturer´s instructions without genomic DNA digestion. RNA samples were quantified using a Nanodrop spectrophotometer (ThermoFisher). 10 μL first-strand cDNA was synthesized using 500 ng RNA with iScript gDNA Clear cDNA Synthesis Kit (Bio-Rad, USA) following the manufacturer´s instructions. Quantitative RT-PCR solution was prepared using the Hieff qPCR SYBR Green Master Mix (Yeason, China), and the reaction was performed using a CFX96 Real time system (Bio-Rad, USA). Primer sequences are shown in Table S2.

## RNA interference (RNAi) gene silencing

Constructs for silencing indicated gene or carrying empty vector (EV) (as control) were expressed in 4-week-old *N. benthamiana* plants in same leaf, side by side, using Agrobacteria infiltration with an $OD_{600} = 0.5$. Four discs were collected at 4 or 8 days post-infiltration to detect silencing efficiency using RT-PCR.

## Virus-induced gene silencing (VIGS)

To silence *NbPtr1*, virus-induced gene silencing was performed as described before [17,25]. In brief, Agrobacteria expressing pTRV1 and pTRV2-Ptr1 were mix in a ratio of 1:1 with a

final dose of $OD_{600} = 1$ and co infiltrated in 3-week-old *N. benthamiana* plants. pTRV2 empty vector was co expressed with pTRV1-EV as negative control. pTRV1-EV mixed with pTRV2-PDS was used as a control to indicate the silenced leaves. Similarly, VIGS of *NbSGT1* was performed as previously described [25] to enable efficient *NbSGT1* silencing without affecting subsequent Agrobacterium-mediated expression. Plant tissues were collected at 9 days post-inoculation to detect silencing efficiency using RT-PCR.

### *R. solanacearum* growth in *N. benthamiana*

The growth of *R. solanacearum* Y45 strain in *N. benthamiana* leaves has been previously described in detail [39,40]. Briefly, *N. benthamiana* leaves expressing the indicated genes were infiltrated with a bacterial suspension containing $10^5$ CFU mL$^{-1}$. Plant tissues were collected 2 days after inoculation for bacterial quantification [40]. Data analysis and representation were performed using Graphpad 7.0 software.

### FRET-FLIM assays

Förster resonance energy transfer – fluorescence lifetime imaging (FRET-FLIM) experiments were performed as previously described [41,42]. Briefly, donor proteins (fused to eGFP) were expressed in a pCAMBIA1300-GFP vector, and acceptor proteins (fused to eRFP) were expressed from vector pGWB554 or pGWB2. FRET-FLIM experiments were performed on a Leica TCS SMD FLCS confocal microscope excitation with WLL (white light laser) and emission collected by a SMD SPAD (single photon-sensitive avalanche photodiodes) detector. Leaf discs of *N. benthamiana* plants transiently coexpressing donor and acceptor, as indicated in the figures, were visualized 30–36 hours after agroinfiltration. Accumulation of the GFP- and RFP-tagged proteins was estimated before measuring lifetime. The tuneable WLL set at 488 nm with a pulsed frequency of 40 MHz was used for excitation, and emission was detected using SMD GFP/RFP Filter Cube (with GFP: 500–550 nm). The fluorescence lifetime shown in the figures corresponding to the average fluorescence lifetime of the donor was collected and analyzed by PicoQuant SymphoTime software. Mean lifetimes are presented as mean ± SEM from at least three independent experiments.

## Supporting information

**S1 Fig. RipE1 is phosphorylated in plant cells.** (A) Phosphorylated peptides detected after immunoprecipitation of RipE1-GFP in *N. benthamiana* leaves followed by LC-MS/MS analysis. The number of the residues, peptide sequences, and mascot ion scores are shown. Phosphorylated residues are shown in red. (B) Representative mass spectra of the phosphorylated peptides shown in (A).
(PDF)

**S2 Fig. RipE1 phosphorylation counteracts its ubiquitination.** (A) Immunoprecipitation assay to determine the ubiquitination status of RipE1 and phosphodeficient mutants. Agrobacterium carrying the indicated constructs were infiltrated as in Fig 1H. Samples were collected 30 hpi, before the appearance of cell death. Given the different RipE1 variants show different protein accumulation, different volumes of the protein samples were loaded to show a comparable RipE1-GFP protein accumulation between different lanes after immunoprecipitation, allowing the detection of ubiquitination in the same amount of RipE1 protein. The relative loading volumes and protein abundance are indicated above lanes. Anti-GFP beads were used for immunoprecipitation. An anti-ubiquitin (P4D1) antibody was used to detect ubiquitinated proteins. Protein marker sizes are shown for reference. This experiment was repeated 3 times, and the quantification of the different repeats is shown in (B). (B)

Quantification of the relative protein ubiquitination of the different repeats of the assay shown in (A), measured using Image J. Ubiquitination values were normalized using the respective protein accumulation and represented as relative to the GFP control for each repeat. Values indicate mean ± SE (n = 3 biological replicates). Different letters indicate significant differences (one-way ANOVA, Tukey's test, p < 0.05). P values are shown for reference. (C) Composite data representation of all the replicates shown in Figs 1H and S2A. Values indicate mean ± SE (n = 6 biological replicates). Different letters indicate significant differences (one-way ANOVA, Tukey's test, p < 0.05).
(PDF)

**S3 Fig. RipE1 and NbUCH15 interact and partially co-localize in the cell periphery and nucleus.** (A) Confocal microscopy images showing the subcellular localization of free GFP and RipE1-GFP. RipE1-GFP or GFP (as control) were expressed in 4-week-old *N. benthamiana* using Agrobacterium (OD$_{600}$ = 0.5). Microscopy images were captured 48 hours post-inoculation. A size bar (25 µM) is shown for reference. The right panel shows a western blot to verify the accumulation of these proteins, ruling out significant GFP cleavage in the RipE1-GFP samples. Blots were analyzed using an anti-GFP antibody, and protein marker sizes are shown for reference. (B) Phylogenetic tree showing the NbUCH proteins identified in this work, together with all the other proteins annotated as UCH in the *N. benthamiana* proteome, showing at least 50% identity when compared with NbUCH05, NbUCH12, and NbUCH15. The tree includes also the proteins encoded by the closest Arabidopsis orthologs, namely AtUBP12 and AtUBP13. (C) Confocal microscopy images showing the subcellular localization of RipE1 and NbUCH15 in *N. benthamiana*. RipE1-RFP or RFP (as control) were co expressed with NbUCH15-GFP or GFP (as control) using Agrobacterium (final OD$_{600}$ = 0.5). Microscopy images were captured 46 hours post-inoculation. Sequential scanning was used to avoid interference between the GFP and RFP channels. A size bar (10 µM) is shown for reference. Each experiment was repeated 3 with similar results. The right panels show the quantification of the fluorescent signals in the boxed areas. (D) Western blot to verify the accumulation of the proteins in (C). Blots were analyzed using an anti-GFP and anti-RFP antibodies, and protein marker sizes are shown for reference. (E) Western blot to verify the accumulation of the proteins in the FRET-FLIM experiments shown in Fig 2C. Blots were analyzed using an anti-GFP and anti-RFP antibodies, and protein marker sizes are shown for reference. An anti-actin antibody was used to verify equal loading.
(PDF)

**S4 Fig. Validation of the silencing efficiency in RNAi and VIGS assays.** (A) Quantitative RT-PCR to determine the expression of *NbUCH05*, *NbUCH12*, and *NbUCH15* in the experiments shown in Fig 3. Samples were taken 4 or 8 dpi. Expression values are relative to the expression of the housekeeping gene *NbEF1a*. Values indicate mean ± SE (n = 3 biological replicates). (B) Quantitative RT-PCR to determine the expression (silencing efficiency) of *NbPtr1*. Samples were taken 8 dpi. Expression values are relative to the expression of the housekeeping gene *NbEF1a*. Values indicate mean ± SE (n = 3 biological replicates). Each experiment was repeat at least three times with similar results.
(PDF)

**S5 Fig. The expression of *RipE1* or *RipAA* is not significantly altered by silencing of *NbUCH* genes.** (A) Quantitative RT-PCR to determine the expression of *RipE1* and *RipAA* in *N. benthamiana* tissues in the experiment shown in Fig 5A. Expression values are relative to the expression of the housekeeping gene *NbEF1a*. Values indicate mean ± SE (n =9 biological replicates). Composite data from 3 independent biological replicates.
(PDF)

**S6 Fig. RipE1 ubiquitination is enhanced upon silencing of *NbUCH15*.** (A) Immunoprecipitation assay to determine the ubiquitination status of RipE1 upon silencing of *NbUCH15*. Agrobacterium carrying the indicated constructs were infiltrated as in Fig 6A. Samples were collected 30 hpi, before the appearance of cell death. In this case, the same sample volumes were loaded into each lane, showing a reduced accumulation of RipE1, but nevertheless a stronger ubiquitination. Anti-GFP beads were used for immunoprecipitation. An anti-ubiquitin (P4D1) antibody was used to detect ubiquitinated proteins. The accumulation of native NbUCH proteins was detected using a custom anti-NbUCH antibody. This experiment was repeated 3 times, and the quantification of the different repeats is shown in (B). (B) Quantification of the relative protein ubiquitination of the different repeats of the assay shown in (A), measured using Image J. Ubiquitination values were normalized using the respective protein accumulation and represented as relative to the empty vector control for each repeat. Values indicate mean ± SE (n = 3 biological replicates). P values are shown for reference according to a Student´s t test. (C) Composite data representation of all the replicates shown in Figs 6A and S6A. Values indicate mean ± SE (n = 6 biological replicates). Asterisk indicates significant differences compared to the control according to a Student's t test (* $p < 0.05$).
(PDF)

**S7 Fig. The phosphorylated residues in RipE1 are conserved among strains belonging to different phylotypes within the *R. solanacearum* species complex.** Protein sequence alignment showing RipE1 versions in different *R. solanacearum* strains. The phosphorylated residues analyzed in this study are indicated in red, together with the conserved domain A, and the catalytic sites.
(PDF)

**S1 Table. Matrix showing the percentage of identity between UCH proteins in *N. benthamiana*.** Matrix showing the percentage of identity between NbUCH proteins identified in this work, together with all the other proteins annotated as UCH in the *N. benthamiana* proteome, showing at least 50% identity when compared with NbUCH05, NbUCH12, and NbUCH15. The matrix includes also the proteins encoded by the closest Arabidopsis orthologs, namely AtUBP12 and AtUBP13. Empty rows correspond to comparisons without a significant identity value.
(XLSX)

**S2 Table. Primers used in this study.**
(DOCX)

**S1 Dataset. Numeric values used to generate all the graphs presented in the figures.**
(XLSX)

## Acknowledgments

We thank Rosa Lozano-Duran for critical reading of this manuscript, Xinyu Jian and Fangyuan Wu for technical and administrative assistance during this work, and all the members of the Macho and Lozano-Duran laboratories for helpful discussions. We thank the PSC Cell Biology and Proteomics core facilities for assistance with confocal microscopy and mass spectrometry.

## Author contributions

**Conceptualization:** Wenjia Yu, Meng Li, Alberto P. Macho.

**Data curation:** Alberto P. Macho.

**Formal analysis:** Wenjia Yu, Meng Li, Alberto P. Macho.

**Funding acquisition:** Alberto P. Macho.

**Investigation:** Wenjia Yu, Meng Li, Wenjun Wang, Haiyan Zhuang, Jiamin Luo, Yuying Sang.

**Methodology:** Wenjia Yu, Meng Li, Wenjun Wang, Haiyan Zhuang.

**Project administration:** Alberto P. Macho.

**Resources:** Cecile Segonzac.

**Supervision:** Alberto P Macho.

**Writing – original draft:** Alberto P. Macho.

**Writing – review & editing:** Wenjia Yu, Meng Li, Cecile Segonzac, Alberto P. Macho.

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
