## [Decision Letter · Decision Letter 0]

29 Oct 2024

PPATHOGENS-D-24-02014A bacterial type III effector hijacks plant ubiquitin proteases to evade degradationPLOS Pathogens Dear Dr. Macho, Thank you for submitting your manuscript to PLOS Pathogens. After careful consideration, we feel that it has merit but does not fully meet PLOS Pathogens's publication criteria as it currently stands. Therefore, we invite you to submit a revised version of the manuscript that addresses the points raised during the review process. Please submit your revised manuscript within 30 days Dec 28 2024 11:59PM. If you will need more time than this to complete your revisions, please reply to this message or contact the journal office at plospathogens@plos.org. Please include the following items when submitting your revised manuscript:* A rebuttal letter that responds to each point raised by the editor and reviewer(s). You should upload this letter as a separate file labeled 'Response to Reviewers '. This file does not need to include responses to any formatting updates and technical items listed in the 'Journal Requirements' section below.* A marked-up copy of your manuscript that highlights changes made to the original version. You should upload this as a separate file labeled 'Revised Manuscript with Track Changes '.* An unmarked version of your revised paper without tracked changes. You should upload this as a separate file labeled 'Manuscript '. If you would like to make changes to your financial disclosure, competing interests statement, or data availability statement, please make these updates within the submission form at the time of resubmission. Guidelines for resubmitting your figure files are available below the reviewer comments at the end of this letter. We look forward to receiving your revised manuscript. Kind regards, Mariana SchusterGuest EditorPLOS Pathogens Bart ThommaSection EditorPLOS Pathogens Michael Malim

Editor-in-Chief

PLOS Pathogens

orcid.org/0000-0002-7699-2064 **Journal Requirements:** **Additional Editor Comments (if provided):** Dear Alberto Macho,

Many thanks for submitting your manuscript on RipE1 stability. This works highlights a new aspect of co-evolution between plants and pathogens and is therefore interesting and relevant for the community.

I agree with the reviewers that this is a well written robust manuscript and consider that addressing their minor suggestions would indeed help polishing the manuscript. Finally, seconding reviewer II, I also suggest to include data on the degradation of phosphomimic versions of RipE1 as this would add robustness to the results.

Best regards,

Mariana Schuster**Reviewers' Comments:** Reviewer's Responses to Questions

**Part I - Summary**

Reviewer #1: This is a very well-designed study that provides important information for understanding the action of the RipE1 effector in the plant cell, and more generally on the control of the stability of these effector proteins, which has been little reported yet for plant pathogens. I have no concerns about the experimental approach used or the interpretation of the results. I found the article well-written and have no details to point out for editing the text.

Overall, it's solid work that should be of interest to the community.

Reviewer #2: In this manuscript, the authors first identified the phosphorylation sites of RipE1 and found that phosphorylation site mutants were more readily degraded in non-host plant Nicotiana benthamiana than wild-type RipE1. This degradation is ubiquitination-dependent, indicating that RipE1 is phosphorylated in plant cells and is less likely to be ubiquitinated.

Next, proteomics analysis of RipE1-binding proteins in N. benthamiana identified NbUCHs, deubiquitinating enzymes, and RNAi silencing of NbUCHs resulted in HR-like cell death. Based on these results, the authors hypothesized that RipE1 targets NbUCH, causing activation of the R protein NbPtr1 and consequently inducing HR. However, silencing of NbPtr1 suppressed RipE1-dependent HR, but silencing of NbPtr1 failed to suppress NbUCHs silencing-dependent HR. These results suggest that NbUCHs are not a target of RipE1 but function to stabilize RipE1 by promoting its deubiquitination.

The RipE1 5A mutant, a phosphorylation site mutant of RipE1, is highly ubiquitinated and degraded, leading to the hypothesis that phosphorylation enhances the interaction with NbUCHs. However, the RipE1 5A mutant interacted with NbUCHs as well as the wild-type RipE1. Based on these results, the authors conclude that the suppression of degradation by phosphorylation of RipE1 and its interaction with NbUCHs are independent mechanisms.

The manuscript is well-written, and the results and conclusions drawn in this study are valid. Although there is some question as to how the phosphorylation of RipE1 escapes the degradation system by host ubiquitination, this manuscript provides very valuable information in terms of understanding the functional regulation of pathogen effectors.

**Part II – Major Issues: Key Experiments Required for Acceptance**

Reviewer #1: None identified.

Reviewer #2: RipE1 5A is considered to be structurally unstable and thus more susceptible to ubiquitination in the Discussion, but how would a phosphorylation mimetic mutation in RipE1 (e.g., RipE1 5D or 5E) affect RipE1 degradation? This experiment is not strictly necessary, but depending on the results, it may help to improve the conclusions of this manuscript.

**Part III – Minor Issues: Editorial and Data Presentation Modifications**

Reviewer #1: My only comment, which is quite minor, concerns the speculation developed in the discussion about the reason for this stability of RipE1, which is only considered to benefit the pathogen. Could we conceive that RipE1's action on its virulence targets is only transient, and that an increase in RipE1 stability through phosphorylation or de-ubiquitination would favor its detection by Ptr1 to lead to HR? In terms of plant defense, this scenario would be more advantageous (as it would result in a stronger response) than a 'simple' degradation of ubiquitinated RipE1 by the proteasome. Related to this, I find Figure 8 not very enlightening for the reader, merely recapitulating the data without providing a truly explanatory or speculative vision.

Reviewer #2: None in particular.

PLOS authors have the option to publish the peer review history of their article (what does this mean? ). If published, this will include your full peer review and any attached files.

**Do you want your identity to be public for this peer review?** For information about this choice, including consent withdrawal, please see our Privacy Policy .

Reviewer #1: No

Reviewer #2: No

 **Figure resubmission:** While revising your submission, please upload your figure files to the Preflight Analysis and Conversion Engine (PACE) digital diagnostic tool, https://pacev2.apexcovantage.com/ . PACE helps ensure that figures meet PLOS requirements. To use PACE, you must first register as a user. Registration is free. Then, login and navigate to the UPLOAD tab, where you will find detailed instructions on how to use the tool. If you encounter any issues or have any questions when using PACE, please email PLOS at figures@plos.org. Please note that Supporting Information files do not need this step. If there are other versions of figure files still present in your submission file inventory at resubmission, please replace them with the PACE-processed versions. **Reproducibility:** To enhance the reproducibility of your results, we recommend that authors of applicable studies deposit laboratory protocols in protocols.io, where a protocol can be assigned its own identifier (DOI) such that it can be cited independently in the future. Additionally, PLOS ONE offers an option to publish peer-reviewed clinical study protocols. Read more information on sharing protocols at https://plos.org/protocols?utm_medium=editorial-email&utm_source=authorletters&utm_campaign=protocols

---

## [Decision Letter · Decision Letter 1]

29 Dec 2024

PPATHOGENS-D-24-02014R1

A bacterial type III effector hijacks plant ubiquitin proteases to evade degradation

PLOS Pathogens

Dear Dr. Macho,

Thank you for submitting your manuscript to PLOS Pathogens. After careful consideration, we feel that it has merit but does not fully meet PLOS Pathogens's publication criteria as it currently stands. Therefore, we invite you to submit a revised version of the manuscript that addresses the points raised during the review process.

Please submit your revised manuscript within 30 days Feb 27 2025 11:59PM. If you will need more time than this to complete your revisions, please reply to this message or contact the journal office at plospathogens@plos.org. Please include the following items when submitting your revised manuscript:

We look forward to receiving your revised manuscript.

Kind regards,

Mariana Schuster

Guest Editor

PLOS Pathogens

Bart Thomma

Section Editor

PLOS Pathogens

Sumita Bhaduri-McIntosh

Editor-in-Chief

PLOS Pathogens

orcid.org/0000-0003-2946-9497

Michael Malim

Editor-in-Chief

PLOS Pathogens

orcid.org/0000-0002-7699-2064

**Additional Editor Comments:**

Dear Alberto Macho,

Thanks for submitting a revised version of your manuscript. I agree with the reviewers that the new version is much improved. Nevertheless, I would like to ask you to please follow the suggestion of one of the reviewers of including the data of RipE1 5D in Figure 1. Also, please keep figure 8 as this graphic summary of the results will likely contribute to their dissemination.

Best regards

Mariana Schuster

**Journal Requirements:**

1) We have noticed that you have uploaded Supporting Information files, but you have not included a complete list of legends. Please include the legend of (Dataset S1) file in the Supporting Information legends after the references list.

2) Please ensure that the funders and grant numbers match between the Financial Disclosure field and the Funding Information tab in your submission form. Note that the funders must be provided in the same order in both places as well. Currently, the order of the funders is different in both places.

Please indicate by return email the full and correct funding information for your study and confirm the order in which funding contributions should appear. Please be sure to indicate whether the funders played any role in the study design, data collection and analysis, decision to publish, or preparation of the manuscript.

**Reviewers' Comments:**

Reviewer's Responses to Questions

**Part I - Summary**

Reviewer #1: The authors have thoroughly addressed the reviewers' comments, significantly improving the manuscript and meeting the conditions for acceptance. Regarding Figure 8, while I remain unconvinced of its scientific value as previously noted in my initial review, I defer to the editor's judgment on whether to retain or remove the figure from the main article.

Reviewer #2: The authors have carefully and thoroughly revised the manuscript in response to the previous peer review comments and have provided clear responses to the concerns raised. The revised manuscript is worthy of acceptance.

**Part II – Major Issues: Key Experiments Required for Acceptance**

Reviewer #1: (No Response)

Reviewer #2: None in particular

**Part III – Minor Issues: Editorial and Data Presentation Modifications**

Reviewer #1: (No Response)

Reviewer #2: The data for the newly added phosphorylated mutant RipE1 (RipE1 5D) is considered to be important for this manuscript, so it should be included in Figure 1 rather than in the supporting figures.

PLOS authors have the option to publish the peer review history of their article (what does this mean? ). If published, this will include your full peer review and any attached files.

**Do you want your identity to be public for this peer review?** For information about this choice, including consent withdrawal, please see our Privacy Policy .

Reviewer #1: No

Reviewer #2: No

**Figure resubmission:**

While revising your submission, please upload your figure files to the Preflight Analysis and Conversion Engine (PACE) digital diagnostic tool, https://pacev2.apexcovantage.com/ . PACE helps ensure that figures meet PLOS requirements. To use PACE, you must first register as a user. Registration is free. Then, login and navigate to the UPLOAD tab, where you will find detailed instructions on how to use the tool. If you encounter any issues or have any questions when using PACE, please email PLOS at figures@plos.org. Please note that Supporting Information files do not need this step. If there are other versions of figure files still present in your submission file inventory at resubmission, please replace them with the PACE-processed versions.
---

## [Editor Report · Decision Letter 2]

4 Jan 2025

Dear Dr. Macho,

We are pleased to inform you that your manuscript 'A bacterial type III effector hijacks plant ubiquitin proteases to evade degradation' has been provisionally accepted for publication in PLOS Pathogens.

Best regards,

Mariana Schuster

Guest Editor

PLOS Pathogens

Bart Thomma

Section Editor

PLOS Pathogens

Sumita Bhaduri-McIntosh

Editor-in-Chief

PLOS Pathogens

orcid.org/0000-0003-2946-9497

Michael Malim

Editor-in-Chief

PLOS Pathogens

orcid.org/0000-0002-7699-2064
---

## [Editor Report · Acceptance letter]

Dear Dr. Macho,

We are delighted to inform you that your manuscript, "A bacterial type III effector hijacks plant ubiquitin proteases to evade degradation," has been formally accepted for publication in PLOS Pathogens.

Best regards,

Sumita Bhaduri-McIntosh

Editor-in-Chief

PLOS Pathogens

orcid.org/0000-0003-2946-9497

Michael Malim

Editor-in-Chief

PLOS Pathogens

orcid.org/0000-0002-7699-2064